# A protocol for a multi-site cohort study to evaluate child and adolescent mental health service transformation in England using the i-THRIVE model

**Moore A.**[1,2,3]*, **Lindley Baron-Cohen K.**[2], **Simes E.**[1,2], **Chen S.**[3], **Fonagy P.**[1,2]

**1** The Anna Freud National Centre for Children and Families, London, United Kingdom, **2** Psychoanalysis Unit, Division of Psychology and Language Sciences, University College London, London, United Kingdom, **3** Department of Psychiatry, University of Cambridge, Cambridge, United Kingdom

* am2708@medschl.cam.ac.uk

**Data Availability Statement:** No datasets were generated or analysed during the current study. All relevant data from this study will be made available upon study completion.

## Abstract

The National i-THRIVE Programme seeks to evaluate the impact of the NHS England-funded whole system transformation on child and adolescent mental health services (CAMHS). This article reports on the design for a model of implementation that has been applied in CAMHS across over 70 areas in England using the 'THRIVE' needs-based principles of care. The implementation protocol in which this model, 'i-THRIVE' (implementing-THRIVE), will be used to evaluate the effectiveness of the THRIVE intervention is reported, together with the evaluation protocol for the process of implementation. To evaluate the effectiveness of i-THRIVE to improve care for children and young people's mental health, a cohort study design will be conducted. N = 10 CAMHS sites that adopt the i-THRIVE model from the start of the NHS England-funded CAMHS transformation will be compared to N = 10 'comparator sites' that choose to use different transformation approaches within the same timeframe. Sites will be matched on population size, urbanicity, funding, level of deprivation and expected prevalence of mental health care needs. To evaluate the process of implementation, a mixed-methods approach will be conducted to explore the moderating effects of context, fidelity, dose, pathway structure and reach on clinical and service level outcomes. This study addresses a unique opportunity to inform the ongoing national transformation of CAMHS with evidence about a popular new model for delivering children and young people's mental health care, as well as a new implementation approach to support whole system transformation. If the outcomes reflect benefit from i-THRIVE, this study has the potential to guide significant improvements in CAMHS by providing a more integrated, needs-led service model that increases access and involvement of patients with services and in the care they receive.

**Funding:** This study is funded by the National Institute for Health Research (NIHR) Applied Research Collaboration (ARC) North Thames (grant number CRJM, 46AZ74-/RCF/CLAHRC/ UCL004, 52AZ95/AA2/UCL5, 52AZ95/AA2/UCL6, UCLP) and UCLPartners, both awarded to AM. The views expressed in this publication are those of the author(s) and not necessarily those of the NIHR, the Department of Health and Social Care or UCLPartners. All research at the Department of Psychiatry in the University of Cambridge is supported by the NIHR Cambridge Biomedical Research Centre (BRC-1215-20014) and NIHR Applied Research Centre. The views expressed are those of the author(s) and not necessarily those of the NIHR or the Department of Health and Social Care.

**Competing interests:** The authors have declared that no competing interests exist.

## Introduction

Since 2015, the NHS has been undergoing the most significant transformation since its foundation in 1948. The Five Year Forward View [1] described a vision for the future of the NHS that focused on new ways of working to improve quality of care by prioritising the needs of service users, rather than of single institutions and structures. This new approach, emphasising collaboration between services and integration of care, aimed to break down traditional barriers between health, social care and voluntary sectors to create a 'whole system-based' approach to service delivery, organised around networks of care. These networks were intended to hold joint accountability for the outcomes of specific population groups, rather than for individual patients [2, 3]. The ambitions of the Five-Year Forward View were translated into a range of NHS England-led programmes which received over £1.6 billion of investment between 2015 and 2020. Child and adolescent mental health services (CAMHS) specifically received £1.25 billion between 2016 and 2021, the most significant funding injection seen for CAMHS to date, to begin a fundamental reform addressing serious concerns surrounding the accessibility and quality of care in CAMHS [2].

This paper describes the development and planned evaluation of the National i-THRIVE Programme that has implemented the THRIVE Framework [4–6] for system change in CAMHS across over 70 areas in England since the beginning of the national CAMHS transformation in 2016. Unlike the tiered service model that CAMHS currently uses, the THRIVE Framework proposes a set of evidence informed concepts and principles that describe an integrated and person-centred care system for children and young people's (CYP) mental health that delivers care according to the individuals' needs, rather than by severity or diagnosis (see S1 and S2 Figs) [4–6].

i-THRIVE, which stands for implementing-THRIVE, has been developed alongside THRIVE to support sites implementing this approach in CAMHS, drawing on implementation science principles [7]. Over 60% of CYP in England currently live in a region who are adopting THRIVE using this approach and who are active members of the i-THRIVE community of practice [8]. This paper reports on the design and protocol of the i-THRIVE approach to implementation, its planned evaluation and the evidence-base it draws from.

## Whole system transformation in the NHS

The predominant NHS England-led improvement programme between 2015 and 2018 focused on developing and evaluating a series of new 'whole system' integrated models of care through pilot sites, known as 'Vanguards' [9]. These were created as locally-driven prototypes for integrating health and social care services within a geographically bound region, aiming to transition the NHS to functioning on a 'place-based' framework. In this model organisations (including acute, community and mental health trusts and local authorities) within a region have joint responsibility for improving the outcomes of the population within the region, rather than for the care delivered to individual patients within their respective organisations. It was anticipated that approaches developed by successful Vanguards would be disseminated across the NHS in England [1, 3, 10].

Aligned to the Vanguard implementation programme were a range of evaluations, which have provided preliminary indications about what could be useful, both in terms of the models and the approaches to implementation used [11–13]. However, due to a range of design and data collection challenges, it has been difficult to draw firm conclusions from them [12, 14–16]. For example, it has been almost impossible to establish which service components included in the Vanguard programme led to improvement, how these affected each organisation, and the subsequent impact on outcomes [12]. Problems stemmed from the rapidity of

implementation and difficulty aligning implementation and evaluation efforts, in particular as evaluation was often reported to be a post hoc activity that was infrequently embedded within the transformation efforts nor prioritised by leadership [11, 12].

Consequently, none of the evaluations were able to measure fidelity to the model, as often there was no clearly ascribed approach at the outset nor any logic models developed to explain the theorised relationship between service changes and outcomes, and there were significant challenges in accessing performance and qualitative data from participating organisations [11, 12]. In contrast, MRC Guidelines highlight that useful evaluation of a complex intervention requires assessment of both the effectiveness of the model being adopted, and the effectiveness of the implementation process itself [17]. Without both, it is difficult to determine if poor outcomes are due to an ineffective intervention, partial implementation, or poor fidelity to the model. Where a model is found to be useful, it is important to understand the factors and processes observed to be associated with improved outcomes. It is also critical that evaluations take place over an appropriate timeframe, as premature evaluation could mislead conclusions about a model's effectiveness, which may reflect incomplete implementation rather than an ineffective model [18].

## Service transformation within CAMHS

During this period of NHS transformation, focus has been placed on improving CAMHS services, supported by the largest injection of spending in its history. Identifying optimal care models and addressing the implementation issues related to whole system transformation for CAMHS has never been more important. Long waiting times have been a major barrier to CYP accessing mental health care in the UK. In 2017 it was estimated that more than 12.5% of CYP in England experience a mental health problem [19]. Prior to the start of national CAMHS transformation, in 2013 the average waiting time to access a routine care appointment in CAMHS was as high as 15 weeks, with only 31% of CYP who required intervention accessing services [20]. For those CYP that were offered support, a lack of flexibility in service models to provide care in line with patient needs and preferences resulted in dissatisfaction in the care received, leading to poor patient engagement, and poor clinical outcomes [5]. Difficulty accessing care was made worse by patients 'falling through the cracks' when transferring between services, as a result of poor staff coordination and organisations not being incentivised to provide care along whole pathways.

A growing body of evidence incorporated into the THRIVE framework indicates that these problems within CAMHS could be improved by actively involving CYP in their own mental health care as collaborative participants and decision-makers, rather than passive consumers of treatment. Enhancing these opportunities through shared decision-making (SDM) is associated with better clinical outcomes as well as greater patient satisfaction [21–23]. Reports from CYP, professionals and carers indicate that multi-agency working improves treatment experience, and promotes a more comprehensive delivery of care [24, 25]. Considering this and additional evidence it is recognized that tackling CAMHS waiting times requires three fundamental changes in practice: (1) tailoring interventions and therapies to fit clinical practice better [26, 27], (2) tailoring interventions and therapies to fit CYP's needs and preferences [27], and (3) building and tailoring interventions and therapies to fit non-clinical (non-NHS) contexts [28]. Building on this evidence, care delivered through goal-focused approaches have been shown to improve patient flow through the service, reducing waiting times, and improving service accessibility [29, 30].

The i-THRIVE implementation programme supports sites to incorporate many of these principles, which are outlined in the THRIVE Framework [4–6], into their delivery of care.

However, while elements are evidence informed, there remains no large-scale evaluation of the effectiveness of the approach in terms of the outcomes achieved, the key components of a service that has adopted the THRIVE principles, nor the optimal approach to its implementation. The national CAMHS transformation, which includes all CCGs from 2016–2022, has over seventy sites that chose to use i-THRIVE as the basis of their transformation efforts. This provides a unique opportunity to undertake a national evaluation of i-THRIVE, and if adequate quality data are obtained, a study of the implementation has the potential to extend knowledge of *what works* in the development of whole system, place-based approaches to delivering care in CAMHS and for the NHS more widely.

## Developing the model of implementation

The i-THRIVE model of implementation takes an evidenced-informed approach to whole system service transformation and has been designed to implement THRIVE's framework of concepts and principles in a CAMHS setting. The THRIVE Framework [4–6] comprises a set of concepts and principles that set out what a 'THRIVE-like' system of CAMHS might look like across the macro (systemic: commissioning and provider leadership systems), meso (organisational structures and relationships, and how these work together within clinical pathways) and micro (individual practitioners' ways of working) dimensions of a mental health care system for CYP (see S1 Table). The i-THRIVE approach to implementation translates these concepts and principles into a series of components developed by drawing on the evidence-base to create a structure and set of tools to guide implementation efforts. The six main components of the i-THRIVE model are described in Table 1, along with the implementation science principles relating to integrated care systems that they draw on. For example, drawing on Normalisation Processing Theory [31, 32], the 'THRIVE Assessment Tool' (see S3 Fig) is designed to provide a practical guide for implementation and a means to assess a site's alignment with THRIVE principles, and the 'i-THRIVE Academy' (see S2 Table) is designed to offer training to staff around ways of working that support the delivery of the THRIVE principles.

With the aim to strike a balance between supporting local determinism and retaining fidelity to the THRIVE Framework, the components of the i-THRIVE model are designed to support sites to deliver care that is aligned to the THRIVE principles while also compatible with the local context, consistent with evidence indicating that innovative service models are more easily adopted if they can fit local contexts [17]. Following this approach, the components of the i-THRIVE model do not aim to prescribe a set protocol for how sites should deliver the THRIVE principles of care, but aim to guide each site to develop a holistic approach to their local interpretation of the THRIVE principles–their own 'THRIVE model of care'–and support them to develop an implementation plan to guide them in how to implement it. For example, SDM is a core THRIVE principle. i-THRIVE does not prescribe *how* shared decision making should be implemented, but provides a range of options and tools. i-THRIVE Option Grids© have been developed by the implementation team [48], however, the i-THRIVE implementation programme has also collated a range of other approaches to SDM that have been successfully used by i-THRIVE sites. Core to i-THRIVE is to make such support resources readily accessible and the i-THRIVE Community of Practice website signposts to all of these along with a slew of relevant resources. The Community of Practice day focussing on SDM includes presentations from a range of sites on how they have approached SDM. The i-THRIVE academy module on SDM, includes training for Option Grids and other approaches. The i-THRIVE Assessment Tool assesses whether SDM has been successfully implemented, and does not prescribe a particular approach. Together, this enables sites to choose an approach that best fits their local context [49].

**Table 1. The components of the i-THRIVE programme.**

| i-THRIVE component | Vectors to guide implementation |
|---|---|
| The i-THRIVE Community of Practice creates a community of sites actively implementing THRIVE across the UK. Shared learning events are held several times per year, offering opportunities to learn from other sites implementing THRIVE, the THRIVE authors and the i-THRIVE Implementation Support Team. | This is designed to:<br>• Strengthen connections between sectors by removing service provision boundaries and creating a support network for sites implementing THRIVE [33–37]<br>• Actively encourage positive communication and team-building across the whole place-based system [37, 38]<br>• Offer a shared environment to support cross-sector working and learning to help build relationships, create a shared vision of care and associated goals [34, 35, 37, 39–41]<br>• To consistently hear from experts in the field of CAMHS transformation |
| The i-THRIVE Implementation Support Team is responsible for overseeing the i-THRIVE Programme, including managing the Community of Practice and i-THRIVE Academy, and developing the i-THRIVE Toolkit. The Team also provides one-to-one support to local services implementing THRIVE. | This is designed to provide:<br>• Active and consistent leadership [36, 41]<br>• A central team working specifically to strengthen the connections between sectors [34, 38]<br>• A clear vision of care and plan for implementation to establish commitment and involvement across all sectors [34]<br>• Supervision for data collection of shared outcomes that prioritise patients at each of the macro, meso and micro levels of care [39]<br>• Adequate resources for cross-sector provision, while ensuring these are distributed equitably across sites [39] |
| The i-THRIVE Approach to Implementation is a four-phase approach to implementation developed drawing on the Quality Implementation Framework [41] and Normalisation Process Theory [31]. | This is designed to provide:<br>• A focus on process issues for implementation, including how integration is defined and organised for evaluation [40, 42]<br>• A structured approach to network building to support cross-sector working [35, 37, 40] |
| The i-THRIVE Toolkit provides information, guidance and a resource to support sites implementing THRIVE. Designed to be flexible to adapt to the local context over time, and to support fidelity to the protocol within each locality. | This is designed to:<br>• Provide a practical resource to determine the level and scope of integration required at each site at baseline and across the evaluation [43–45]<br>• Support implementation planning towards a shared vision and outcome of care [34, 39]<br>• Support the development of the service integration rationale and monitor the evaluation of this [46]<br>• Provide data that could be used to inform policy on integrated care [47] |
| The i-THRIVE Academy provides access to coaching and training for front-line staff to support the delivery of the THRIVE principles. The training focuses on the core THRIVE principles and needs-based categories (see S2 Table). | This is designed to:<br>• Ensure all education and resources are patient-centred [35–37]<br>• Offer an opportunity for cross-sector learning and relationship building to support integration and partnership working [35, 37, 38, 40]<br>• Encourage positive communication and attitudes surrounding implementation efforts [38] |
| The i-THRIVE Option Grids are paper-based decision-aids developed as part of the i-THRIVE Toolkit to improve shared decision-making between CYP, their families and staff by facilitating conversations about care, using a shared language. | This is designed to:<br>• Offer a practical resource to support patient-centred treatment and cross-sector collaboration [35–37]<br>• Provide a structured approach for working towards a shared goal to improve clarity when developing and working through these [37, 41] |

The 'i-THRIVE Approach to Implementation' is manualized and is structured to be a four-phase process, drawing on the Quality Implementation Framework [50] (see S2 Fig) and Nor-malisation Process Theory [31, 32], for implementing complex interventions. Each phase is supported by a range of tools, described in detail on the programme website [51]. Phase One includes forming a comprehensive understanding of the local system, including the existing challenges and needs of the local population. Stakeholders and system leaders across health, social care, education and third sector organisations are invited to engage with the programme during this phase, and governance is established. The THRIVE Assessment Tool is used to assess how 'THRIVE-like' the CAMHS system is compared to the THRIVE principles. Through multi-agency discussion, areas of strength and weakness are identified, which sup-ports the development of an implementation plan and a measurable approach for monitoring change. Phase Two includes building capacity within the system by appointing 'THRIVE Champions', and identifying the training needs of managers leading transformation and staff on the front-line. Front-line training is designed to be conducted through the i-THRIVE Academy, created to provide a set of four modules to support staff development, each of which addresses the core requirement of one of the four THRIVE domains (see S2 Table). A local i-THRIVE Community of Practice is also created to provide a mechanism for shared learning across implementation teams. To ensure that the data required to support improvement will be available and used within the teams, data collection mechanisms are established during this phase. Phase Three focuses on implementation of the new system using a variety of change management and quality improvement methods (e.g. 'Plan, Do, Study, Act' cycles), as well as establishing information and quality infrastructures within providers and commissioners. Measurement systems enabling collaborative assessment of progress are set up during this phase to identify potential issues so that these can be tackled across the locality. Phase Four focuses on learning, embedding and sustaining changes to ensure these become 'business as usual' once the transformation programme is complete. This involves reflection on what is working better or has been learnt from implementation, and offers an opportunity to provide feedback and share local learning with the broader i-THRIVE Community of Practice.

Taking each of these components into account, a logic model has been created for the pro-gramme and used as a framework for evaluation (see Fig 1).

## Evaluation objectives

The evaluation has two sets of objectives, the first involves evaluation of the effectiveness of the i-THRIVE model in improving the performance of CAMHS services, and the second relates to evaluating the approach to implementation.

### Evaluating the effectiveness of the i-THRIVE model

Evaluation of i-THRIVE model focusses on the relationship between the model of care used and the outcomes achieved within a region. Our *research questions* are: (1) Does i-THRIVE lead to improvements in patient experience, clinical outcomes and service performance out-comes in CYP's mental health services? (2) Which elements of the service lead to improve-ments in outcomes? (3) Who benefits from i-THRIVE, and what are the consequences for equity and diversity?

### Evaluating the process of implementation

The evaluation of the implementation process extends from the logic framework in Fig 1 and draws on MRC guidance to understand context, quantify implementation and explore mecha-nisms of impact (see Fig 2) [17, 18]. The *Research Questions* focus on: (1) What approaches to

| Inputs | Processes | Outputs | Outcomes | Impact |
|---|---|---|---|---|
| *Components needed for successful implementation* | *Actions required to achieve implementation objectives.* | *Implementation program deliverables* | *The changes that implementation aims to deliver* | *Medium-long term effects achieved as a result of implementation* |
| 1. i-THRIVE Approach to implementation and THRIVE principles.

2. Suitable funding *(Health Foundation,* UCL Partners, NHS Innovation Accelerator, Health Education England).

3. Leadership: Multi-agency Partnership Board including: CAMHS, senior organisational leadership, local authority, CCG and iTHRIVE implementation support team.

5. Implementation support in form of dedicated project management and access to local data. | 1. Whole system stakeholder mapping & engagement.

2. Establish local multi-agency 'i-THRIVE Community of Practice' with regular shared learning events.

3. Follow 'i-THRIVE Approach to Implementation'.

4. Staff attend 'i-THRIVE Academy'.

5. Coaching of local site Implementation Leads and Clinical Leads by i-THRIVE Implementation Support Team. | 1. Care delivered according to young people's needs.

2. Staff observe THRIVE principles in practice.

3. Improved multi-agency working.

4. Shared decision making.

5. Effective signposting of young people, carers & professionals to services across the system.

6. Clear treatment endings.

7. Evidence based care provided to specific patient groups.

8. Diverse in modalities of care.

9. Multi-agency risk assessment plans as required.

10. Routine outcome measurement outcomes, used for improving quality fof care. | 1. Improved access to MH support and treatment.

2. Reduced waiting times for CAMHS.

3. Improved experience of care.

4. Reduced episodes of care.

5. Reduced length of stay.

6. Improved multi-agency working.

7. Improved staff satisfation.

8. Validated approach to implementation.

9. Improved knowledge & capacity within NHS for service transformation. | 1. Improved clinical, social, education and economic outcomes.

2. Effective method of integrated service transformation available for wide use and adaptation to other settings.

3. Greater equity of care.

4. Better staff retention.

5. More efficient use of public funds. |

**Fig 1. i-THRIVE logic model.** The i-THRIVE logic model shows the inputs, processes, outputs and the expected outcomes and impact of implementation.

implementation have been employed by sites? (2) What is the relationship between implementation (including measures of context, fidelity, dose and clinical pathways) and outcomes (including patient experience, service performance and clinical outcomes)? (3) What are the barriers and facilitators to implementation?

**Hypotheses.** We predict that using an evidenced-informed approach to implementation, i-THRIVE, will lead to improved fidelity to the THRIVE model across macro, meso and micro systems. This will lead to more integrated services and the barriers to implementation will be more easily overcome. Collectively, we hypothesise, this will lead to improved service and patient outcomes, in particular improved access to services, shorter waiting times and shorter length of stay. By providing needs-based care and access to services according to patients' preferences, we expect better engagement with services, improved experience of care, fewer dropouts and better clinical outcomes. Shared decision making and improved signposting will support broader access, positively impacting on diversity and inclusion in services.

## Methods

### Study setting and design

We plan to use a matched cohort study design. Twenty CAMHS sites across England are to be included in the study, comprising: ten 'accelerator sites' have been identified that adopted the

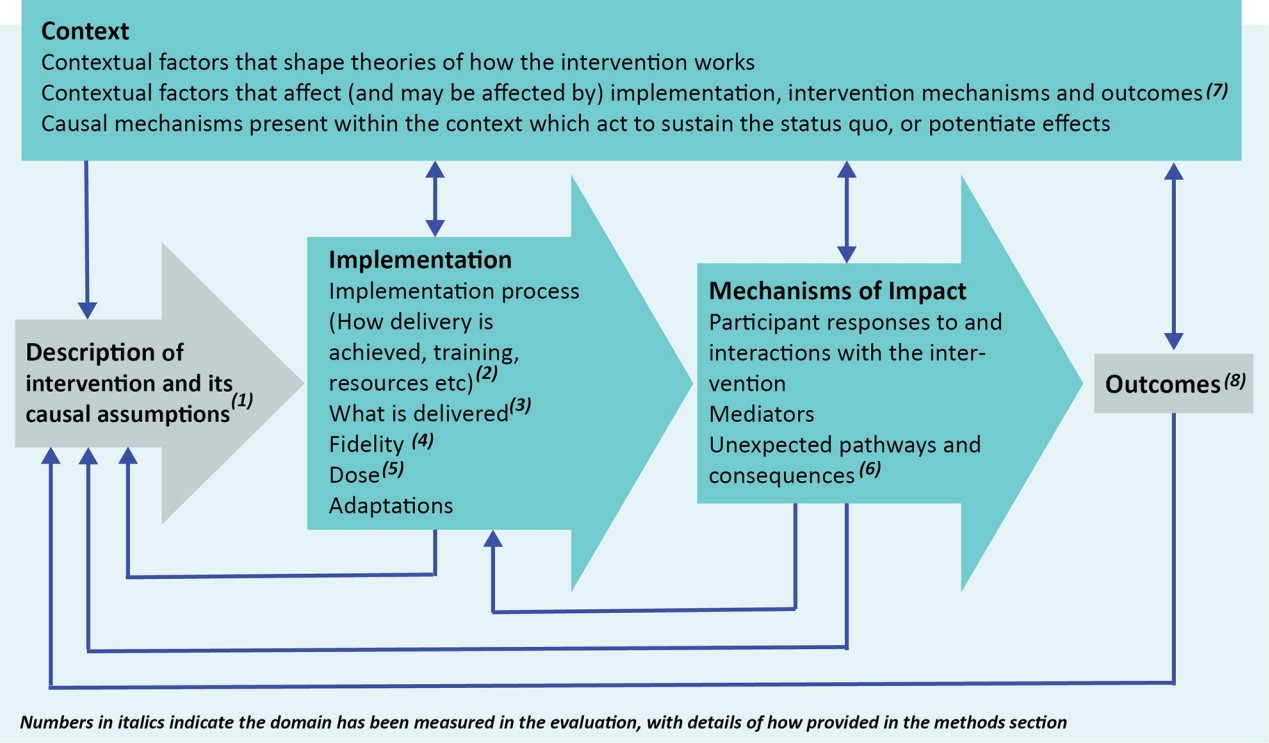

**Fig 2. The conceptual model underpinning the study design (Moore et al., 2015), indicating measurement domains.**

i-THRIVE model from the start of the NHS England-funded CAMHS transformation. We identified ten 'comparator sites' that chose to use different transformation approaches within the same timeframe which were comparable on broad demographic criteria. All sites are defined as geographically discrete CAMHS commissioning areas within England, aligned to either a single NHS Clinical Commissioning Group (CCG) or a small number of CCGs working in collaboration, with responsibility for commissioning CAMHS for their local population. Since the i-THRIVE model aims to support a whole-system integrated approach to delivering CYP's mental health care, each site will be described in terms of the CCG(s), the trust(s) providing CAMHS, and the partner Local Authority(ies).

Site recruitment is now complete. Sites were recruited through an email invitation to participate in the study sent to all CCGs in the UK inviting them to participate in an implementation programme to support the adoption of the i-THRIVE model. Sites applied to be an implementation site and were asked to demonstrate commitment to implement the model over four years from both health and social care, and to participate in the evaluation. In return they were offered implementation support via the i-THRIVE Implementation program and the funded evaluation. Ten sites met the criteria and were adopted into the study. It was important to identify matched control as closely as possible by baseline characteristics, commitment to transformation and approach to joint working between health and social care. Our approach was to match for baseline characteristics first (population size and density reported by the Office of National Statistics–Mid 2015 Population Estimates for Clinical Commissioning Groups in England [52], combined CAMHS and CCG funding 2016–2017 and level of deprivation defined by the Department for Communities and Local Government data—English Indices of Deprivation 2015 [53]. An ordered list of preferred sites for each test site was

comprised and each was approached in order of preference according to matching criteria. Sites were then interviewed to establish the commitment of health and social care to CAMHS transformation and participation in the evaluation over four years. Sites were asked to agree not to implement i-THRIVE within the four years of the evaluation. If all the criteria were met, controls were included in the study until we had ten matched controls.

## Data collection

Data will be collected between 2015 and 2020 for all sites. The measurement plan is embedded within the logic model (see S3 Table for an overview of all measures to be collected). Covid-19 has delayed data collection which has not been completed and data analysis remains to be performed.

## Outcome measures

Patient and service level data collection will take place at baseline and then annually over the four-year study period. Outcome measures are to be categorised into three groups:

Clinical outcomes will comprise measures of symptom improvement and patient reported outcomes that are routinely recorded by each site.

Patient experience and engagement with services is to be assessed using routine patient experience measures provided by each site (such as The Friends and Families Test, and Children's Global Assessment Scale) and engagement through the proportion of patients attending appointments compared to those who do not.

Service performance outcomes are focused on three areas: *access*, *waiting times*, and *efficiency*. Access will be measured by: the number of new referrals received to the service, the number of referrals assessed by triage, and by Tier 3 services and the number of patients receiving treatment. Waiting times will be assessed by calculating three indicators: the average waiting times between referral received to triage, referral received and assessment, referral received and the first episode of care within a specialist service and the maximum waiting time. Efficiency will be measured by the average number of: contacts per patient, face-to-face appointments, non-face-to-face appointments, discharges, and re-referrals. Sub-analyses will be conducted to assess ethnicity, age and diagnoses to assess the impact of on equity and diversity.

Fig 3 and Table 2 show the type of data to be collected, define the measures to be used, and how these measures relate to the CAMHS care pathway.

Each of the measures is numbered and corresponds to the measure of the same number found in Fig 3.

## Process evaluation measures

The approach to data collection for the process evaluation has been designed based on the MRC guidance on how to evaluate the process of implementation for a complex intervention. Fig 2 illustrates the conceptual model, based on Moore et al [18], showing the domains measured within this evaluation. In summary, we measure context (details of the local context plus barriers and facilitators to implementation), fidelity (how close local models align with THRIVE principles), dose (the quantity of what is implemented), reach (the extent that the intervention reaches its target audience), and pathway mapping (details of the structures of child mental health pathways, the services offered and the extent to which these are integrated).

Context will be explored to understand the barriers and facilitators to implementation through a survey based on the inner and outer setting constructs defined by the Consolidated

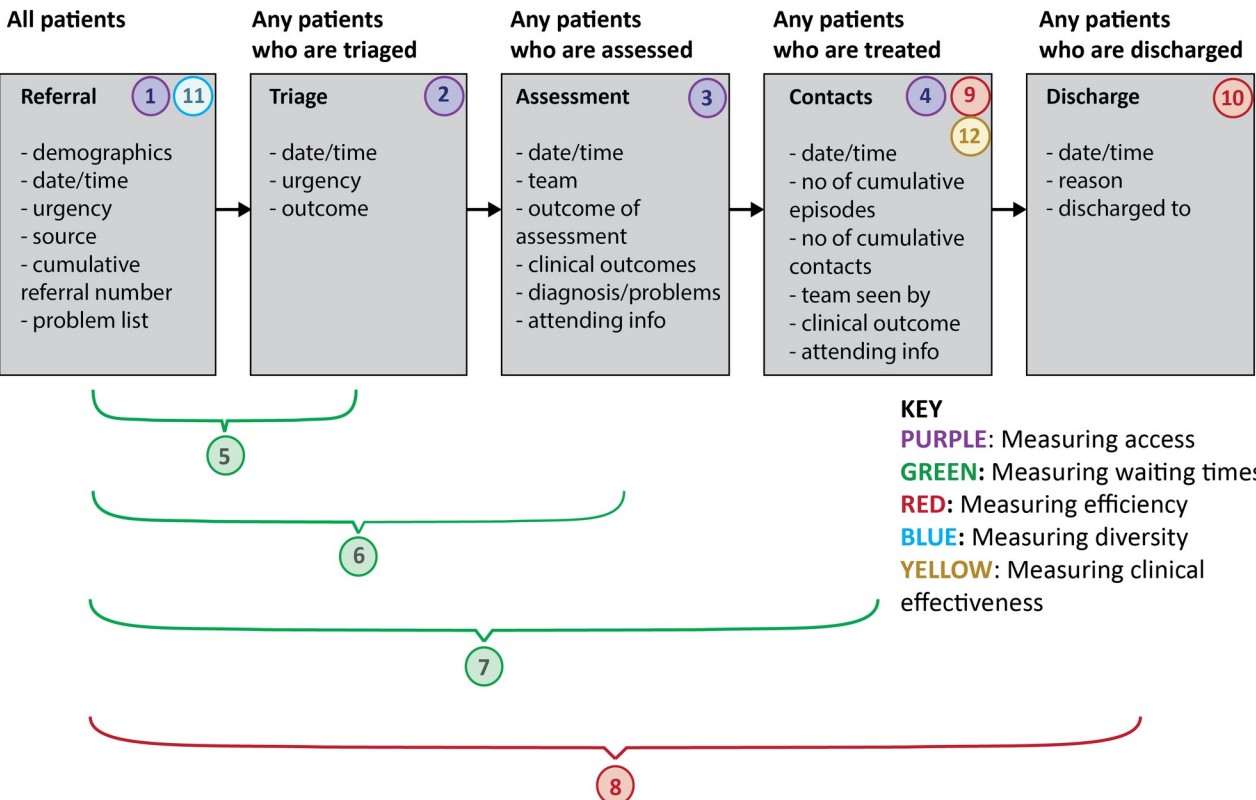

**Fig 3. Quantitative measures collected per patient, defining 'access', 'diversity', 'waiting times', 'efficiency' and 'clinical outcomes'.** The CAMHS pathway is illustrated by the grey boxes. Our base cohort includes all patients referred to a site within the four-year period of the evaluation. For every patient referred we will collect demographic information (age at referral, ethnicity and presence of learning disability). For each subsequent contact point (triage to discharge) the information shown in the relevant box will be collected. This refers to one episode of care. In the case of a patient being re-referred, each subsequent referral will be identified as a separate episode of care. Episodes of care will be recorded accumulatively and given an ID unique to the patient in question. A range of measures have been constructed for the evaluation using data from different parts of the data model. These are numbered 1–12 above and details are provided in Table 2.

Framework for Implementation Research (CFIR) [54]. This is a validated, mixed-method, multi-level framework that conceptualises implementation according to five areas and 39 constructs. The survey will be completed by the i-THRIVE Implementation Leads for accelerator sites, and the CAMHS Transformation Leads for comparator sites.

Fidelity will be assessed by the alignment of services to the THRIVE principles across the macro (senior system leadership and commissioning), meso (service management) and micro (front-line professionals working with CYP) levels of the system. Data will be collected through semi-structured interviews using the i-THRIVE Assessment Tool, a measure that has been developed for this specific purpose, and which will be validated during the course of the i-THRIVE programme. Purposive sampling will be used to recruit three interviewees per system level at each site, to collect a multi-agency perspective across CAMHS, third sector, clinical commissioning groups, education and local authority services. Interviews will be conducted at two time-points: baseline and follow-up (up to four years after service implementation), and will be digitally recorded and transcribed verbatim. The interview transcripts will be scored by researchers who are blind to data collection in relation to how 'THRIVE-like' the service described in the transcript is performing.

Dose and reach will be assessed using a nine-item 'adoption' survey that has been developed based on the RE-AIM framework [55] which is to be sent via email to all front-line staff

**Table 2. Description and definition of the outcome measures and the source of data to be used for the calculation.**

| Construct | Source of data | Method of calculation |
|---|---|---|
| **Access** | | |
| 1. Access to CAMHS services | All referrals into a CAMHS service. | i. Total number of referrals to CAMHS within a year<br>ii. Sources of referrals |
| 2. Access to CAMHS services | All CYP who are triaged by the service. | i. The proportion of cases that are triaged (total triaged/total referrals) |
| 3. Access to CAMHS assessment | All CYP assessments carried out by a service. | i. The proportion of cases that are assessed (total assessed/total referrals) |
| 4. Access to CAMHS treatment | Data relating to the first treatment/intervention session. | i. The proportion of cases that lead to treatment (total CYP receiving at least one intervention session/total referrals) |
| **Waiting times** | | |
| 5. Mean waiting time for triage | Date of referral and date of triage. | i. Mean difference in time between referral and triage |
| 6. Mean wait for assessment | Date of referral and date of assessment. | i. Mean difference in time between referral and assessment |
| 7. Mean wait for treatment | Date of referral and date of assessment. | i. Mean difference in time between referral and first treatment session |
| **Efficiency** | | |
| 8. Mean length of stay | Date of referral and date of discharge. | i. Mean difference in time between referral and discharge date |
| 9. Patterns of service use | Data relating to contacts with CYP or caregivers as part of treatment. | i. Mean and mode number of sessions per episode of care<br>ii. Proportion face to face and non-face to face contacts<br>iii. Proportion of 'did not attends' (including sub-analysis of reasons)<br>iv. Mean of episodes of care per patient |
| 10. Number of discharges | All discharges from CAMHS. | i. Total number of discharges<br>ii. Proportion of discharges (total discharges/total accepted into treatment)<br>iii. Reasons for discharge |
| **Diversity** | | |
| 11. Diversity of CYP accessing CAMHS | All referrals into a CAMHS service. | i. Ethnic diversity of CAMHS referrals<br>ii. Proportion of referrals with learning disability<br>iii. Age range<br>iv. Urgency of referrals<br>v. Presenting problems |
| **Clinical Outcomes** | | |
| 12. Clinical outcomes within episodes of care | Clinical outcome measures completed as part of an assessment, intervention or discharge. | i. Mean baseline need/severity, as measured by site<br>ii. Mean change in need/severity within an episode of care<br>iii. Mean change in need/severity by patient (over multiple episodes of care)<br>iv. Mean experience of care |

working within the accelerator sites, to measure their understanding of i-THRIVE and the THRIVE principles. The survey is to be sent at two time-points: baseline and follow-up (four years after service implementation). In addition, mechanisms of impact of service transformation will be explored through semi-structured interviews with both accelerator and comparator sites through questions related to barriers and facilitators of the service implementation.

Pathway mapping will be undertaken to compare the structure of CAMHS pathways at baseline and after CAMHS transformation to explore whether transformation led to pathways becoming more consistent with NHS England guidelines and the extent to which services and pathways are integrated. We will record: the services provided in a region, including NHS, local authority, third sector and school based mental health interventions; who provides the service, its modality and its relationship to other services in the pathway; and the number of access and assessment points in each system. Data will be collected by reviewing documents of local transformation plans at baseline and after implementation. These will then be supplemented by semi-structured interviews with the site leads to confirm key details and the accuracy of the maps.

We will gain a clear view of what is implemented in place of THRIVE within control sites by bringing together pathway data to understand the changes to the structure of clinical pathways, through the interviews with sites (approximately 18 per site which provide the detail of how sites approached transformation at macro, meso and micro level), and through a survey of the implementation leads.

## Ethics approval and consent to participate

The NHS/University Joint Research Office reviewer the study and determined it to be a service evaluation and therefore does not require ethics or review by the REC (IRAS application number: 250439). For qualitative aspects, participants are implementation leads, managers and front-line staff, and are interviewed about their service transformation project. Verbal informed consent to take part is obtained at the beginning of each interview, including explicit consent for the interview to be recorded, analysed and used as part of the service evaluation. This is recorded and transcripts have been made of each recording to document this. The data used for the quantitative evaluation is retrospective routinely collected service data extracted from electronic records. It is anonymised at source by business intelligence staff at each trust, removing all identifying information, and after review by local information governance leads, is provided to the evaluation team. Consent for the use of retrospective de-identified service data for evaluation purposes is not legally or ethically required and the requirement for ethics was waived by the joint research office.

## Data analysis plan

The qualitative data will be coded, sorted and classified following the 'framework approach' based on the CFIR framework. General statistical test such as t-tests, ANOVA tests, and chi-square tests will be used to evaluate group differences between accelerator and comparator sites, or between the baseline and follow up data collection time points. Difference-in-difference (DID) or Interrupted Time Series (ITS) models will be adopted to test the causal relationship between the i-THRIVE intervention, and service and patient outcomes. The DID design is recommended for quasi-experimental studies that compare the outcomes of groups exposed to different policies and environmental factors at different times [56]. We will compare trusts over time which could have experienced i-THRIVE with the comparison trusts. Care will be taken that the comparison trusts do not become THRIVE-like using our specially developed metric of intervention fidelity to control for so-called "spillover" effects. In order to strengthen

the comparison group we intend to extend the DID design with propensity scores weighting that will enable us to include many more potential control observations from the NHS data set from sites that are more similar to those in the treated group [57]. A key assumptions of DID designs is common trends assumption, that observations should differ by the same amount in every period [56] suggesting that divergences from the common trend after the policy intervention are due to the implementation of i-THRIVE and not to other confounding factors. This assumption will be tested in regression models controlling for other potential confounders such as trust size, the level of economic deprivation, etc. Structural equational modelling will be conducted to test the causal relationship of potential variables to determine which elements of the service lead to improvements in outcomes. Poisson regression with a population offset (to model a rate rather than a count) will be used for variables relating to access to treatment, and where the outcome of interest is a probability (e.g. patterns of service use including 'did not attends') logistic or Probit regression will be chosen as model options. To obtain correct statistical inference (standard errors), repeated observations over time will also be taken into account [58]. All qualitative data will be analysed using NVivo (version 12) and all quantitative data will be analysed using R (version 4.0.0).

## Data management plan

Data collected will be qualitative and quantitative in nature. Anonymised interviews and service performance data will be stored securely on password protected UCL servers for the duration of the study and then for a further ten years. Anonymised data sets may be transferred to University of Cambridge for analysis. Transfer will be via secure file transfer using the encrypted UCL dropbox account and will be stored on password protected Department of Psychiatry, University of Cambridge servers. Only members of the research team will have access to the data. In line with UCL data management policy, following publication all data will be archived at the university and then destroyed after ten years.

## Discussion

This paper describes the development and planned evaluation of the National i-THRIVE Programme, which includes a model for implementation that has been applied in CAMHS, and a protocol for their evaluation. The collection and evaluation of clinical records is currently under way and the results will be reported in subsequent papers. This programme offers an exciting opportunity to closely observe the implementation of a national transformation of CAMHS [1, 7]. As noted above, considerable investment has been made in children and young people's mental health (CYP MH) (over £900m in additional funding is being made available during the period covering the first five years of the Long-Term Plan on top of existing mental health spend). The additional funding is there to support, among other initiatives, the growth in accessibility of CAMHS and CYP MH crisis services as well as the continued expansion of specialist CYP MH community eating disorder services to meet the access standard and the implementation of Mental Health Support Teams in schools and colleges. Our study overlaps with this increased Government commitment to CYP MH enabling us to explore if the implementation of an evidence informed framework for meeting growing need for CYP MH services can support, guide and improve the introduction of new services enabled by fresh resources. We anticipate that the i-THRIVE framework will facilitate and effectively support the whole system transformation demanded of CYP MH services. The THRIVE model was designed to differentiate levels of clinical need enabling a more efficient distribution of resources at a time of substantial increase in the demand for services driven by growing prevalence and greater willingness of YP to recognize and request support for MH problems

[20, 59–61]. We expect that training managers and frontline workers in the THRIVE principles along with other arms of the implementation programme of i-THRIVE will bring measurable micro, meso and macro benefits in terms of quality, efficiency and outcomes in the delivery of MH care to CYP and families.

Past efforts of evidencing the benefit of whole system health care transformation have not invariably met with success. Recently, the UK Government reported plans to establish whole system approaches as a legal requirement for NHS services [62]. Two major NHS policy changes present an opportunity to approach mental healthcare in a new way: Integrated Care Systems (ICS) and the Community Mental Health Framework (CMHF) transformation programme. Both emphasise integration and collaboration across teams, requiring a new approach to performance measurement and decision-making. The challenge of achieving integrated care has highlighted the importance of conceptualising and understanding the complex interdependencies within local mental healthcare systems [63]. Monitoring care by subpopulation, team, and service will be crucial for ICSs concerned with population health management. In mental health, although there is general agreement that this is a positive way forward, given that the largest evaluation of whole system approaches to delivering care in the NHS through the Vanguard New Care Programme was unable to provide clear support for this due to serious implementation challenges, it is critical that further understanding about the implementation of these approaches is established, and what factors lead to implementing successful transformations before expecting that any model could be sustainable [2, 64].

In the Vanguard New Care Programme, despite significant transformation efforts in many cases, implementation challenges meant that full transformation was not accounted for in most evaluations, with the majority of study follow-up periods lasting for no more than a year, which was not long enough to demonstrate meaningful effects on outcomes [15, 16]. There were also limited mechanisms in place for the transformation and evaluation teams to meet to coordinate data collection, or to share local knowledge that could be relevant to the analysis and reporting of service outcomes [12]. This meant that implementation efforts could not benefit from the learning generated, and obstructed the development of relationships and the trust required to enable partners to 'let go of control' of service operations, which were essential for effective cross-boundary working at both individual and organisational levels [12, 65]. Together, these challenges led to significant delays for service transformation to operationalise [66], and it has been almost impossible to establish which service components led to improvement, how these affected each organisation, and the subsequent impact on outcomes [12]. Consequently, we have still not been able to draw conclusions about the impact of these whole system transformations in the New Care Programme, nor about the best way to implement them.

Mental healthcare in general, including CYP MH, is arguably best understood as a complex, adaptive system of interrelated services, organisations, and actors [67]. Outcomes are dependent on relationships between these entities; studying facets in isolation can be misleading [68, 69]. Complexity in the system also requires contending with 'radical uncertainty'—appreciating that not everything can be predicted [70]. Local health systems have begun using routine data for population health management [71], but a missing element has been the involvement of service users and staff to agree local healthcare goals, identify and prioritise metrics, and interpret data [72]. The i-THRIVE Approach to Implementation, that addresses each of these practical and operational issues to whole system transformation may be of value for future service implementation teams to consider adopting [73, 74]. However, until there is more evidence for implementing whole system approaches to delivering care, and between services in the NHS specifically, it may be prudent for policy-makers remain cautious and consider what

they hope to achieve, and how they plan to evaluate the success of their policy, before establishing this [42].

The initiatives that the New Care Models introduced are still important. A significant change is needed in the culture of how CYP are perceived by the services that support them. This was strongly reflected in the recommendations of the Health and Social Care Committee's Expert Panel which noted limitations with the balance of competencies in the mental health workforce and pointed to the absence of good local leadership and management to ensure that national ambitions to improve mental healthcare are met. Integrating services and professionals working with CYP and their families, as well as information across disciplines, should improve the quality of care provided [2, 75]. However, for implementation of such approaches to be successful, integration needs to be carefully defined from the start, to ensure the management of implementation, as well as the outcomes intended, are properly planned and can translate into viable governance forms. Too often, terms such as 'whole system' and 'integrated care' have been used in a general sense, and described through principally conceptual models that lack clarity surrounding the specific integration intended and how it will be organised [42]. This is particularly important in relation to organising accountability across the system-wide framework [2], and to ensure policy-makers can make use of available data determining the level and scope of integration, for any hope of translating successful implementation from one context to another [43, 45, 76]. The recent Health Select Committee Report on CYP MH [77] recommends that Government departments, local government and the health system act together to promote CYP MH to prevent new crises emerging and proposes a Cabinet sub-committee to bring together different departments to make sure system wide coordination happens. The i-THRIVE approach we have outlined presents the best available framework yet for implementing the intended transformation and should ensure that the impact of transformation across each of the key domains, at the macro, meso and micro levels of the system, envisioned by the Select Committee's sub-committee, are kept in focus and offer a broad perspective of the impact of an evaluation for improving CYP's mental health care, which has so far been limited in previous evaluations.

In addition, to successfully integrate services that may differ considerably at the start of transformation in terms of organisational culture and structure, expertise is required [78, 79]. This comes from providing strong leadership at both the clinical and system-wide level, as well as appropriate timeframes for all involved to build new relationships, and learn (as well as unlearn) routines [75]. Each of these components benefit implementation and facilitate a positive culture for change. Given the complexities of service transformation, it is essential that each of these resources are invested before any improvements in organisational efficiency may be visible [42]. The implementation efforts described in this paper are intended to establish the foundations for a learning health care system [80, 81], both at system level guided by patient data and organizational level showing structures, processes, and culture that promote the potential for continuous improvement based on internal learning. i-THRIVE is designed to establish a culture open as much to learning from the monitoring of internal experience as from external published research. The present evaluation is intended to create a model for integrating qualitative and quantitative data from multiple sources to solve problems in design and execution in the future and test and modify new approaches rapidly in order to put insight into action [82]. The i-THRIVE model that incorporates each of these components required to support the whole system, at local levels and across multiple organisational boundaries, may therefore potentially provide an effective protocol for future service transformation teams to adopt to establish systemic change. The spirit of the initiative is for the ability to learn is embedded in the structure of CYP community mental health organisations and their internal processes at

every level, and reinforced by the i-THRIVE culture and the leadership and staff behaviour promoted in its communities of practice.

The process of transformation in CAMHS is ongoing and the evaluation of this implementation model will be reported separately. It is of hope that, in addition to the ambitious evaluation design of the model, by also closely following MRC guidelines to conduct a complete assessment of the process of implementation, the results of this research will be able to provide a comprehensive understanding of the impact of this approach across multiple environments, from which the findings can meaningfully inform the ongoing transformation of CAMHS across different regions of the country.

## Supporting information

**S1 Fig. The THRIVE framework needs-based categories and corresponding inputs (figure from Wolpert et al 2019).** A figure and accompanying text providing a description of the THRIVE framework.
(PDF)

**S2 Fig. Illustrating the i-THRIVE approach to implementation, based on Meyers et al 2012.** A figure illustrating the i-THRIVE Approach to illustration together with the details of where to find additional information on the website.
(TIF)

**S3 Fig. The Thrive assessment tool.** A figure of the Thrive assessment tool that will be used to measure the fidelity in each of the sites.
(PDF)

**S1 Table. Description of how the THRIVE principles translate to the macro, meso and micro system levels.** A table that describes the THRIVE principles for each of the three system levels.
(TIF)

**S2 Table. Description of i-THRIVE Academy learning modules.** A table describing each of the four i-THRIVE Academy Modules, and how they relate to the THRIVE framework.
(TIF)

**S3 Table. The i-THRIVE measurement plan embedded within the logic model.** Showing the logic model item in the first column, the approach to measurement and the corresponding section of how this is approached in the protocol. A table showing how each aspect of the logic model has been translated into an outcome measure, and where the description can be found in the protocol.
(TIF)

## Acknowledgments

The authors wish to acknowledge their indebtedness to the clinicians, managers and academics who have created THRIVE and their generous help and thoughtful advice throughout the development of i-THRIVE and the evaluation protocol to assess its effectiveness. In particular Miranda Wolpert, Rachel James, Peter Fuggle, Rita Harris, Sally Hodges, Duncan Law, Andy Wiener, Ann York, Peter Fonagy, Simon Munk, Caroline McKenna, Melanie Jones and Isobel Flemming. We would also like to thank IIse Lee, Bethan Morris, Nkasi Stoll, Meghan Davis, Sophie Howell, Oliver Owrid, Rudolf Cardinal and Lida Efstathopoulou for their help with the research.

## Author Contributions

**Conceptualization:** Moore A., Fonagy P.

**Data curation:** Simes E.

**Formal analysis:** Moore A., Chen S., Fonagy P.

**Funding acquisition:** Moore A., Simes E., Fonagy P.

**Methodology:** Moore A., Chen S., Fonagy P.

**Project administration:** Simes E.

**Resources:** Moore A.

**Supervision:** Moore A., Fonagy P.

**Visualization:** Moore A.

**Writing – original draft:** Moore A., Lindley Baron-Cohen K., Fonagy P.

**Writing – review & editing:** Simes E., Chen S.

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
