## [Decision Letter · Decision Letter 0]

15 Jun 2022

PONE-D-22-03202Protocol for a multi-site case control study evaluating child and adolescent mental health service transformation in England using the i-THRIVE modelPLOS ONE

Dear Dr. Moore,

Thank you for submitting your manuscript to PLOS ONE. After careful consideration, we feel that it has merit but does not fully meet PLOS ONE’s publication criteria as it currently stands. Therefore, we invite you to submit a revised version of the manuscript that addresses the points raised during the review process. Please consider the concerns by one reviewer. 

We look forward to receiving your revised manuscript.

Kind regards,

Andrew Soundy

Academic Editor

PLOS ONE

Journal Requirements:

Reviewers' comments:

Reviewer's Responses to Questions

**Comments to the Author**

1. Does the manuscript provide a valid rationale for the proposed study, with clearly identified and justified research questions?

Reviewer #1: Yes

Reviewer #2: Yes

2. Is the protocol technically sound and planned in a manner that will lead to a meaningful outcome and allow testing the stated hypotheses?

Reviewer #1: Partly

Reviewer #2: Yes

3. Is the methodology feasible and described in sufficient detail to allow the work to be replicable?

Reviewer #1: No

Reviewer #2: Yes

4. Have the authors described where all data underlying the findings will be made available when the study is complete?

Reviewer #1: Yes

Reviewer #2: Yes

5. Is the manuscript presented in an intelligible fashion and written in standard English?

Reviewer #1: Yes

Reviewer #2: Yes

6. Review Comments to the Author

You may also provide optional suggestions and comments to authors that they might find helpful in planning their study.

Reviewer #1: The study protocol has hugely ambitious scope and has the potential to directly influence services for CYP mental health and also how implementation science is undertaken.

Major queries:

How were/will the THRIVE intervention and non-THRIVE control sites be selected? Understanding what might drive differences in THRIVE/control sites at baseline is central to the evaluation and is not sufficiently well described.

There are two key considerations; i) how you select the comparison group, ii) how you understand what was implemented in place of THRIVE. Both are critical for the evaluation.

If the control sites have not yet been selected then it would be helpful to describe the data that will be used to identify sites with similar characteristics (size, urbanicity, deprivation etc). If these are available at national level then please consider running several comparison approaches in parallel for at least some aspects of the evaluation – for example using synthetic control groups that have similar patterns of service use before the start of the study.

Minor points:

Lines 87/88: given how widespread the coverage of the THRIVE intervention is reported to be, can you add some further information about how sites opted in (or out) of this model.

Lines 92-126: I agree that place-based care is increasingly important (e.g. given the shift towards integrated care systems), however, I felt that it wasn’t clear which services were viewed as being part of this system. Because of this it is hard to gauge whether the outcomes of interest have wide enough breadth – e.g. knock effects on CYP mental health presentations to A&E. Some more information about which services this relates to would be helpful either here or in the section on Service Transformation within CAMHS.

Lines 230-241: I am unclear how data collection works at sites where THRIVE has not been implemented.

Table 2: not entirely clear how some of these will be measured and compared – e.g. clinical outcomes mean baseline need/severity, as measured by site.

Line 350-398: is it likely that some control sites may be quite “THRIVE-like”? Will this change what you do if this is the case?

Line 416-425: please consider using a wider range of comparison groups and methods for the quantitative outcomes (if possible)

Reviewer #2: This is a very thorough protocol for an ambitious and impressive study. The only section that I thought may benefit from more detail (or reference to more detail reported elsewhere if applicable) was the planned data analysis section.

7. PLOS authors have the option to publish the peer review history of their article (what does this mean?). If published, this will include your full peer review and any attached files.

Reviewer #1: **Yes: **Ruth Blackburn

Reviewer #2: No

---

## [Author Response · Author response to Decision Letter 0]

3 Aug 2022

Journal Requirements:

We have amended the formatting to be in line with the guidelines. 

We intended to say that the data would not be available until the end of the study (which will be after this protocol is published). This may not have been clear enough in our section on Data Access and so we have re-drafted this to provide additional clarity. 

We have refreshed the reference list and as far as we are aware, we have not cited any papers that have been retracted. 

Responses to reviewers: 

Reviewer #1: The study protocol has hugely ambitious scope and has the potential to directly influence services for CYP mental health and also how implementation science is undertaken.

Thank you for this supportive recognition of the scope and ambition we have for this project. 

Major queries:

How were/will the THRIVE intervention and non-THRIVE control sites be selected? Understanding what might drive differences in THRIVE/control sites at baseline is central to the evaluation and is not sufficiently well described.

The sites have been selected. We have included additional information about the process of selection for both controls and test sites (see section ‘study design and setting’). 

There are two key considerations; i) how you select the comparison group, ii) how you understand what was implemented in place of THRIVE. Both are critical for the evaluation. If the control sites have not yet been selected then it would be helpful to describe the data that will be used to identify sites with similar characteristics (size, urbanicity, deprivation etc). If these are available at national level then please consider running several comparison approaches in parallel for at least some aspects of the evaluation – for example using synthetic control groups that have similar patterns of service use before the start of the study.

We agree that these are both critical points – thank you for highlighting these. We have aimed to provide clarification on our approach in response to this. We have provided detail about the approach to selection of the comparison group in the section ‘study design and setting’. We recognise the need to match the sites as closely as possible in terms of baseline characteristics, as well as commitment to transformation and approach to joint working. To address this, we matched initially on baseline characteristics, and then interviewed preferred sites to find sites motivated to transform CAMHS and take a partnership approach between health and social care. We also required them to be willing to participate in the evaluation for four years and to commit not to implement i-THRIVE within this time period. Part of our initial analysis will be to measure the fidelity to THRIVE at baseline. We will then be able to determine the extent to which this changes over the implementation period. We will report this in the first results paper. 

Minor points:

Lines 87/88: given how widespread the coverage of the THRIVE intervention is reported to be, can you add some further information about how sites opted in (or out) of this model.

This detail has been added. 

Lines 92-126: I agree that place-based care is increasingly important (e.g. given the shift towards integrated care systems), however, I felt that it wasn’t clear which services were viewed as being part of this system. Because of this it is hard to gauge whether the outcomes of interest have wide enough breadth – e.g. knock effects on CYP mental health presentations to A&E. Some more information about which services this relates to would be helpful either here or in the section on Service Transformation within CAMHS.

We have included detail of how we understand integrated care systems for the purpose of this study. 

Lines 230-241: I am unclear how data collection works at sites where THRIVE has not been implemented.

This section describes the approach to implementation in i-THRIVE sites rather than the data collection approach. We collected data in the same way in both control and test sites: interviews, surveys and use of de-identified routinely collected CAMHS data. The same processes and approach was used in all sites. 

Table 2: not entirely clear how some of these will be measured and compared – e.g. clinical outcomes mean baseline need/severity, as measured by site.

At the moment we are not able to set this out, as we have not been able to collect the data from all sites. Most sites use CGAS and so we expect this to be the most likely outcome measure that will be used. We will be able to develop the measurement approach in detail once the data has been collected and mapped across the sites. 

Line 350-398: is it likely that some control sites may be quite “THRIVE-like”? Will this change what you do if this is the case?

We don’t think so. We hope that at base line the controls and test sites are similarly THRIVE-like, as this will be a good indicator that the matching process was successful. Our assessment will be to determine the change in THRIVE-likeness over the 4 years. If the controls are equally as THRIVE-like, then we are likely to conclude that CAMHS transformation as driven by NHS England leads to similar CAMHS service structure changes regardless of the model and approach to implementation taken. We hypothesise that sites participating in the i-THRIVE Implementation programme will be more THRIVE-like at follow up. And we will then test the relationship between THRIVE-likeness and outcomes. 

Line 416-425: please consider using a wider range of comparison groups and methods for the quantitative outcomes (if possible)

We have significantly expanded the analysis section by including a more detailed description of our plans.

Reviewer #2: This is a very thorough protocol for an ambitious and impressive study. The only section that I thought may benefit from more detail (or reference to more detail reported elsewhere if applicable) was the planned data analysis section.

Thank you for the comments on the protocol. We have significantly expanded the analysis section by including a more detailed description of our plans.

---

## [Decision Letter · Decision Letter 1]

19 Sep 2022

PONE-D-22-03202R1A protocol for a multi-site case control study to evaluate child and adolescent mental health service transformation in England using the i-THRIVE modelPLOS ONE

Dear Dr. Moore,

Thank you for submitting your manuscript to PLOS ONE. After careful consideration, we feel that it has merit but does not fully meet PLOS ONE’s publication criteria as it currently stands. Therefore, we invite you to submit a revised version of the manuscript that addresses the points raised during the review process.

Please submit your revised manuscript by  If you will need more time than this to complete your revisions, please reply to this message or contact the journal office at plosone@plos.org. Please include the following items when submitting your revised manuscript:A rebuttal letter that responds to each point raised by the academic editor and reviewer(s). You should upload this letter as a separate file labeled 'Response to Reviewers'.A marked-up copy of your manuscript that highlights changes made to the original version. You should upload this as a separate file labeled 'Revised Manuscript with Track Changes'.An unmarked version of your revised paper without tracked changes. You should upload this as a separate file labeled 'Manuscript'.If applicable, we recommend that you deposit your laboratory protocols in protocols.io to enhance the reproducibility of your results. Protocols.io assigns your protocol its own identifier (DOI) so that it can be cited independently in the future. For instructions see: https://journals.plos.org/plosone/s/submission-guidelines#loc-laboratory-protocols. Additionally, PLOS ONE offers an option for publishing peer-reviewed Lab Protocol articles, which describe protocols hosted on protocols.io. Read more information on sharing protocols at https://plos.org/protocols?utm_medium=editorial-email&utm_source=authorletters&utm_campaign=protocols.

We look forward to receiving your revised manuscript.

Kind regards,

Retno Asti Werdhani, M.Epid, PhD

Academic Editor

PLOS ONE

Journal Requirements:

Additional Editor Comments:

Adequate matching variables

Adequate qualitative’s variable measurement for mix method approach

Well explained research gap

Well described evaluation indicator (outcome and process measure)

Need to be have the definition of urbanicity, funding, level of deprivation and expected prevalence mental health care needs for matching variables

The calculation of N = 10 for each case and control

The method is not case control. It’s more like a cohort (start with the exposure vs non exposure: I thrive is considered the exposure, and then will be followed up until 4 years after service implementation. If you used secondary data, then the method will be historical cohort

Reviewers' comments:

Reviewer's Responses to Questions

**Comments to the Author**

1. Does the manuscript provide a valid rationale for the proposed study, with clearly identified and justified research questions?

aas.     

Reviewer #1: Yes

Reviewer #2: Yes

2. Is the protocol technically sound and planned in a manner that will lead to a meaningful outcome and allow testing the stated hypotheses?

Reviewer #1: Yes

Reviewer #2: Yes

3. Is the methodology feasible and described in sufficient detail to allow the work to be replicable?

Reviewer #1: Yes

Reviewer #2: Yes

4. Have the authors described where all data underlying the findings will be made available when the study is complete?

Reviewer #1: No

Reviewer #2: Yes

5. Is the manuscript presented in an intelligible fashion and written in standard English?

Reviewer #1: Yes

Reviewer #2: Yes

6. Review Comments to the Author

You may also provide optional suggestions and comments to authors that they might find helpful in planning their study.

Reviewer #1: I am satisfied that the queries previously raised have been addressed. I look forward to seeing the results of the study in the future.

Response to Q4 relating to data availability: the manuscript is a protocol so the requirement to make data available therefore does not apply.

Reviewer #2: I felt the authors addressed all of the issues raised both in their responses and additions to the manuscript. I wish them the best of luck with this project and look forward to reading the results once they are available.

7. PLOS authors have the option to publish the peer review history of their article (what does this mean?). If published, this will include your full peer review and any attached files.

Reviewer #1: **Yes: **Ruth Blackburn

Reviewer #2: No

---

## [Author Response · Author response to Decision Letter 1]

2 Mar 2023

Thank you for your sharing the editors and reviewers’ comments with us. Please find enclosed 

1. A rebuttal letter that responds to each point raised by the academic editor and reviewer(s). 

2. A marked-up copy of your manuscript that highlights changes made to the original version.

3. An unmarked version of your revised paper without tracked changes. 

Journal Requirements: 

We have reviewed the reference list and all 83 references have been checked and updated as required. 

Additional Editor Comments:

2. Need to be have the definition of urbanicity, funding, level of deprivation and expected prevalence mental health care needs for matching variables

We thank the Editor for these comments and have attempted to comply with each. In relation to definitions of matching criteria we have referred to the ONS 2015 Survey which we used as the bases. The following text has been added to the study setting and design section of the manuscript on page 17:

Our approach was to match for baseline characteristics first (population size and density reported by the Office of National Statistics – Mid 2015 Population Estimates for Clinical Commissioning Groups in England, combined CAMHS and CCG funding 2016-2017 and level of deprivation defined by the English Indices of Deprivation 2015.

Reviewers' comments:

3. The calculation of N = 10 for each case and control

The method is not case control. It’s more like a cohort (start with the exposure vs non exposure: I thrive is considered the exposure, and then will be followed up until 4 years after service implementation. If you used secondary data, then the method will be historical cohort. 

Thank you for your feedback. 

We have updated the description of the study design to cohort study. The justification of the number of case and control sites chosen was based on the number of early adopters identified which could be reasonably matched with other sites using different approaches to service transformation that were comparable in size and demographics. The number is also consistent with minimum size of a cluster randomised control trial design (Medical Research Council (2002) Cluster Randomised Trials: Methodological and Ethical Considerations. MRC, London. Available at http://www.mrc.ac.uk/index/publications/pdf-) Cluster trials with fewer than five clusters per arm are considered inadvisable. We attempted to balance out cluster level confounding by matching on key demographics. As the study is a cohort study there are only pragmatic rather than statistical justification for the number of sites selected. 

Our understanding is that because the manuscript is a protocol the requirement to make data available does not apply. 

We are grateful for the opportunity we have been given to improve our manuscript and we have done our best to address all the issues raised. We will be delighted to make any further amendments you feel may be required. We look forward to hearing from you in due course.

---

## [Editor Report · Decision Letter 2]

14 Mar 2023

A protocol for a multi-site cohort study to evaluate child and adolescent mental health service transformation in England using the i-THRIVE model

PONE-D-22-03202R2

Dear Dr. Moore,

We’re pleased to inform you that your manuscript has been judged scientifically suitable for publication and will be formally accepted for publication once it meets all outstanding technical requirements.

Kind regards,

Retno Asti Werdhani, M.Epid, Ph.D

Academic Editor

PLOS ONE
---

## [Editor Report · Acceptance letter]

27 Apr 2023

PONE-D-22-03202R2 

A protocol for a multi-site cohort study to evaluate child and adolescent mental health service transformation in England using the i-THRIVE model 

Dear Dr. A.:

I'm pleased to inform you that your manuscript has been deemed suitable for publication in PLOS ONE. Congratulations! Your manuscript is now with our production department. 

Kind regards, 

on behalf of

Dr. Retno Asti Werdhani 

Academic Editor

PLOS ONE